# Prognostic Effects of Vasomotor Reactivity during Targeted Temperature Management in Post-Cardiac Arrest Patients: A Retrospective Observational Study

**DOI:** 10.3390/jcm10153386

**Published:** 2021-07-30

**Authors:** Mun Hee Choi, Sung Eun Lee, Jun Young Choi, Seong-Joon Lee, Da Sol Kim, Minjung Kathy Chae, Eun Jung Park, Ji Man Hong

**Affiliations:** 1Department of Neurology, School of Medicine, Ajou University, 164 World cup-ro, Yeongtong-gu, Suwon-si 16499, Korea; choimoonhee09@gmail.com (M.H.C.); plumpboy@hanmail.net (S.E.L.); taz312@gmail.com (J.Y.C.); editisan@gmail.com (S.-J.L.); 2Department of Emergency Medicine, School of Medicine, Ajou University, 164 World cup-ro, Yeongtong-gu, Suwon-si 16499, Korea; mutjeo@gmail.com (M.K.C.); amita62@nate.com (E.J.P.); 3Department of Brain Science, School of Medicine, Ajou University, 164 World cup-ro, Yeongtong-gu, Suwon-si 16499, Korea; 4Neuroscience Graduate Program, Department of Biomedical Sciences, School of Medicine, Ajou University, 164 World cup-ro, Yeongtong-gu, Suwon-si 16499, Korea; ekthf0422@naver.com

**Keywords:** out-of-hospital cardiac arrest, prognosis, targeted temperature management, ultrasonography, doppler, transcranial, vasomotor reactivity

## Abstract

Early and precise neurological prognostication without self-fulfilling prophecy is challenging in post-cardiac arrest syndrome (PCAS), particularly during the targeted temperature management (TTM) period. This study aimed to investigate the feasibility of vasomotor reactivity (VMR) using transcranial Doppler (TCD) to determine whether final outcomes of patients with comatose PCAS are predicted. This study included patients who had out-of-hospital cardiac arrest in a tertiary referral hospital over 4 years. The eligible criteria included age ≥18 years, successful return of spontaneous circulation, TTM application, and bedside TCD examination within 72 h. Baseline demographics and multimodal prognostic parameters, including imaging findings, electrophysiological studies, and TCD-VMR parameters, were assessed. The final outcome parameter was cerebral performance category scale (CPC) at 1 month. Potential determinants were compared between good (CPC 1–2) and poor (CPC 3–5) outcome groups. The good outcome group (*n* = 41) (vs. poor (*n* = 117)) showed a higher VMR value (54.4% ± 33.0% vs. 25.1% ± 35.8%, *p* < 0.001). The addition of VMR to conventional prognostic parameters significantly improved the prediction power of good outcomes. This study suggests that TCD-VMR is a useful tool at the bedside to evaluate outcomes of patients with comatose PCAS during the TTM.

## 1. Introduction

Unconsciousness is not only an initial manifestation in post-cardiac arrest syndrome (PCAS), but also a common neurological presentation that persists in intensive care units (ICUs) following resuscitation [1,2]. Brain injury after cardiac arrest is a major factor in determining sustained disability and final mortality of patients with PCAS, which is often terminated by the withdrawal of life-sustaining treatment [3]. Since the introduction of targeted temperature management (TTM), the timing of the prognosis for patients with comatose PCAS has been conservatively delayed to avoid self-fulfilled prophecy with very early withdrawal of life-sustaining treatment [4,5].

Although TTM is a standard therapy to effectively prevent systemic and brain reperfusion injury after cardiac arrest, it may provide us with an empty time box of ‘we do not know the clinical outcome of this period’ [6,7]. Numerous physicians have embraced a multimodal approach to evaluate neurological outcomes after the TTM period (at least 72 h) with clinical examinations, electrophysiological studies, blood biomarkers, and neuroimaging to improve prognostic uncertainty. Recent interests in providing acute neuroprotective interventions have emerged in the era of state-of-the-art neurocritical care [8]. For these reasons, early neurological prognostication (within 72 h) of patients with PCAS using bedside ancillary tests may be promising in the future [4,9].

Cerebral autoregulation (CA) is the ability to constantly maintain the cerebral blood flow despite changes in blood pressure, which is usually regulated by the dilative capacity of the intracerebral arterioles [10,11]. Carbon dioxide (CO_2_) is a potent vasodilator on those vasculatures [12]. Transcranial Doppler (TCD) can be remarkably useful in the ICU because it is a noninvasive and portable device with high temporal resolution [12]. The TCD vasomotor reactivity (VMR) test is a semiquantitative CA test that is used to measure the CA potential of regulating arterioles through CO_2_ inhalation or breath-holding [13,14]. In preclinical and clinical experiments, cerebral ischemia has been related to CA impairments, as sudden cessation of blood flow leads to a dysfunction of cerebral arterioles and capillaries during and after ischemia [15,16]. In this context, several studies have reported that impairment of CA had a high potential to predict poor prognosis in patients with PCAS [17,18]. Therefore, this study aimed to investigate the feasibility of the TCD-VMR test in predicting outcomes following cardiac arrest, especially during the TTM period, in comatose patients.

## 2. Materials and Methods

### 2.1. Study Population

An observational cohort of patients with out-of-hospital cardiac arrest in Ajou University Medical Center from January 2016 to February 2020 was analyzed retrospectively. The study population also included surviving patients with TTM and TCD-VMR test. Patients who aged <18 years, died within 72 h, not treated with TTM, had CA caused by intracerebral mechanism, and had no TCD examination were excluded. TCD-VMR tests were not performed in patients with the presence of intracerebral pathology, refusal to additional examination by caregiver, and condition of hemodynamical instability (Figure 1). This study was approved by the Ethics Committee of Ajou University Medical Center (Approval No. AJIRB-MED-MDB-17-196). Informed consent was waived because of the retrospective nature of this study. All procedures were part of the standard care at our hospital.

### 2.2. TTM and Management

Patients who were comatose despite return of spontaneous circulation (ROSC) were admitted to an ICU. TTM was conducted using temperature-managing devices with a feedback loop system, such as a surface cooling device (Arctic Sun; Bard Medical, Covington, GA, USA) or an intravascular cooling device (Coolgard; Zoll Medical Corporation, Chelmsford, MA, USA). According to the standardized protocol of our institution, which is based on guidelines [19], the temperature was maintained at 32–36 °C, followed by rewarming at a rate of 0.15–0.25 °C/h. General management was performed according to the protocol of Ajou University Medical Center, including adequate control of sedation, analgesia, and shivering. In accordance with our hospital’s protocol, all patients with TTM were given a sedative drug from the first day of ICU admission.

### 2.3. Multimodal Assessment for Neurologic Prognostication

Multimodal assessment for neurologic prognostication was executed in accordance with the methods in a previous study [4]. Clinical examinations included pupillary light reflex, corneal reflex, and Glasgow Coma Scale motor response score. Within 72 h of ROSC, the best response was retained for prognostication. A Glasgow Coma Scale motor response score ≤2 was designated as absent motor response. Non-contrasted brain computed tomography (CT) was performed for screening of intracranial causes of cardiac arrest. Gray-to-white matter ratio (GWR) and bilateral Alberta stroke program early CT score (ASPECTS-b) were measured by two separate investigators (M.H. Choi, J.M. Hong) blinded to the clinical findings, according to a previous study [20]. Electrophysiological examinations, such as electroencephalography (EEG) and somatosensory evoked potential, were performed after 72 h. All patients underwent standard EEG, and a considerable change in the background activity after noxious or photic stimulations was investigated. Bilateral median nerve somatosensory evoked potential detected a presence of cortical N20 wave on both cerebral hemispheres. These findings were interpreted by a certified electrophysiologist (J.Y. Choi) blinded to the clinical findings.

### 2.4. TCD-VMR

The TCD-VMR test was conducted using breath-holding methods [13,14,21,22,23]. During TTM, the TCD was performed using a 2 MHz pulse-wave Doppler machine (Pioneer TC 8080; Viasys Healthcare, Madison, WI, USA) by experienced sonographers for reducing inter-operator variability. Both middle cerebral arteries (MCAs) were insonated and measured at a depth of 50–60 mm with two probes fixed to the patient’s head with a headband. In the case of a poor temporal window, the basilar artery was insonated and measured at a depth of 70–80 mm. Breath-holding was started when the mean flow velocities (MFV) were recorded stably. Breath-holding was not performed when there was a reverberating flow pattern suggesting massively increased intracranial pressure and probable circulatory arrest [24]. The ventilator was disconnected from the patient for approximately 40 s. If an increase in hypercapnia by more than 10% compared to baseline was not achieved, the breath-holding test was repeated. The partial pressure of arterial CO_2_ (PaCO_2_) levels was obtained twice from the arterial line during rest and immediately after the end of breath-holding.

The MFV and pulsatility indices of both MCAs were continuously recorded during rest and breath-holding maneuvers (Figure 2). MFV changes during breath-holding were determined by subtracting the MFV at rest from the MFV after breath-holding. The cerebral VMR (%) was determined by dividing the MFV change by the resting MFV: ({MFV after breath-holding − resting MFV}/resting MFV) × 100 [23].

### 2.5. Outcome Assessment

Neurological outcomes were assessed at 1 month using the Glasgow–Pittsburgh Cerebral Performance Categories (CPC). Patients were dichotomized into good (CPC 1–2) and poor (CPC 3–5) outcome groups. Outcome assessment was established prospectively, and examiners were blinded from the observational cohort of patients with out-of-hospital cardiac arrest.

### 2.6. Statistical Analysis

Clinical, imaging, electrophysiological, and TCD parameters were compared between the good and poor outcome groups. Categorical and continuous variables were analyzed using the chi-square test and Mann–Whitney test, respectively. Statistically significant variables (*p* < 0.05) were selected and incorporated into univariate logistic regression. To identify factors that influence outcome, age, initial rhythm, cause of CA, time to ROSC, and early prognostic parameters, including pupillary light reflex, GCS motor score, S100, CT score, and TCD-VMR, were entered into the multivariable logistic regression model. Receiver operating characteristic (ROC) curves were used to determine and compare the prognostic power of VMR and other imaging tests. The optimal cutoff value of VMR for predicting good neurological outcomes was determined to be the largest value by calculating the Youden J statistic (J = (sensitivity + specificity) − 1). The performance of tests for predicting good outcomes was obtained as the area under the ROC curve. Model performance was also evaluated by calculating C-statics, and the improvement in predictive accuracy was evaluated by calculating integrated discrimination improvement (IDI) and net reclassification improvement (NRI) values [25]. Statistical analyses were conducted using SPSS software version 25.0 (IBM Corp., Armonk, NY, USA) and R software version 4.0.4 (R Project, Vienna, Austria). A *p*-value < 0.05 was considered significant.

## 3. Results

### 3.1. Patient Characteristics

From January 2016 to February 2020, 158 consecutive comatose patients (103 men and 55 women, median age of 60) resuscitated from out-of-hospital cardiac arrest were included in this study. All patients were treated with TTM. Table 1 summarizes patient characteristics and prognostication test parameters. At 1 month, 41 patients (25.9%) showed good neurological recovery (CPC 1–2), and 117 patients (75.1%) had poor outcomes (CPC 3–5). Young age (56 (41–62) vs. 61 (51–73), *p* = 0.015), male sex (78.1% vs. 60.7%, *p* = 0.045), shorter cardiopulmonary resuscitation time (15 (11–26) vs. 26 (17–40), *p* < 0.001), shockable initial rhythm (68.3% vs. 11.1%, *p* < 0.001), and cardiogenic cause of CA (75.6% vs. 24.8%, *p* < 0.001) were more prevalent in the good CPC group than in the poor CPC group. Other comorbidities were not different between the two groups.

In the standard prognostication test, the good CPC group showed traditional results in all parameters. Clinical examinations, including pupillary light reflex, corneal reflex, and motor response, were executed during TTM. The good CPC group was revealed to have a higher frequency of positive responses in all clinical examinations. (no pupillary light reflex bilaterally, 7.3% vs. 55.6%, *p* < 0.001; GCS motor score, 2.6 ± 1.8 vs. 1.5 ± 1.0, *p* < 0.001; GCS motor score ≥ 2, 39.02% vs. 11.97%, *p* < 0.001). Protein S100, a serologic marker that reflects brain injury, was significantly lower in the good outcome group at the initial stage and 24 h later (0.94 (0.46–2.58) vs. 3.85 (2.13–6.74) at the initial stage, *p* < 0.001; 0.11 (0.07–0.18) vs. 2.24 (0.46–7.38) 24 h later, *p* < 0.001). Values of imaging parameters predicting early ischemic change, GWR, and ASPECTS-b [20] were higher in the good CPC group (GWR, 1.23 ± 0.06 vs. 1.16 ± 0.08, *p* < 0.001; ASPECTS-b, 14 (8–18) vs. 5 (2–11), *p* < 0.001). In total, 146 (92.4%) patients were evaluated by EEG, and an unreactive EEG background was more frequent in the poor CPC group than in the good CPC group (56.3 vs. 94.7%; *p* < 0.001). Absence of bilateral N20 was also more prevalent in the poor CPC group (*n* = 0/32, 0.0% vs. *n* = 67/91, 73.6%; *p* < 0.001).

### 3.2. Associations of VMR with Outcome

Across all patients, TCD-VMR tests were conducted during TTM and sedation/analgesia (median, 29.7 h after ROSC). Bilateral poor temporal windows were noted in 14 patients (8.8%), and they were excluded from the TCD parameter analysis. The baseline MFV was higher in the poor CPC group than in the good CPC group, but the pulsatility index was not different. MFV changes during breath-holding presenting the mean VMR were higher in the good CPC group than in the poor CPC group (54.4% ± 33.0% vs. 25.1% ± 35.8%, *p* < 0.001) (Table 1). Twenty-nine patients (18.4%) showed reverberating flow, representing severely increased intracranial pressure and impending brain death. All patients with reverberating flow were allocated into the poor CPC group. The TCD parameters of the age/sex-matched control group are described and compared with the patients with cardiac arrest in the Appendix A.

In the univariate logistic regression, older age, shockable rhythm, cardiogenic cause, shorter time to ROSC, presence of pupillary light reflex, GCS motor score ≥ 2, GWR ≥ 1.17, ASPECTS-b ≥ 7, initial S100 < 2.37 μg/L, and mean VMR ≥ 30.3% were significantly associated with good CPC at 1 month. After adjustment for classic prognostic parameters, the mean VMR remained as a strong independent predictor of outcomes (adjusted odds ratio: 122.80, confidence interval: 8.66–1741.91, *p* < 0.001, Table 2).

### 3.3. Prognostic Performance of VMR for Predicting Good Outcome

Figure 3 shows the ROC curve as an indicator of good CPC having a more powerful prognostic ability when VMR parameters were added. The area under the ROC curve for the prediction of a good CPC was the largest at 0.792 for VMR (cutoff value, 30.3). The NRI and IDI values were calculated to assess the improvement of prediction power for good neurological outcomes using a combination of imaging markers and VMR. The NRI and IDI values show that adding VMR to the traditional imaging findings is advantageous for predicting good neurological outcomes (Table 3). The Appendix A present the prognostic performance of the clinical markers, imaging parameters, biomarkers, and VMR for predicting good outcomes at 1 month.

## 4. Discussion

In this study, compared with potential predictors using other conventional parameters, the TCD-VMR test during the TTM period was a significant prognostic determinant of the final outcomes of patients with comatose PCAS.

Our data show that decreased VMR was related to poor outcomes in 1 month. VMR is the dilative or reserve capacity of intracerebral regulating arterioles measured by changes in CO_2_, and it has been considered a semi-static CA test [13]. Our result shows that the preserved VMR exhibits a good vasomotor response recovered from initially damaged dilative capacity of the cerebral arterioles after cardiac arrest, which is consistent with previous results [17,18]. Indeed, comatose conditions are quite similar to heart attack conditions at the start of TTM; therefore, physicians find it difficult to determine the actual extent of brain damage with the clinical approach alone in patients with PCAS. Some EEG studies have shown prognostic determinants indicating early good brain functions to predict favorable outcomes, but they are either nonspecific or difficult to interpret [2,9]. For this reason, this modality is not easy to apply as a real-time bedside monitoring procedure without EEG experts and facilities. However, the TCD-VMR test is quite intuitive, and vasodilative functions of the cerebral arterioles can be interpreted within 1 min [14]. Therefore, even in patients with comatose PCAS, this method can be easily applied to real-world situations.

In the classic concept of the viability thresholds in the ischemic penumbra, a gradual decrease in cerebral blood flow can lead to a reversible or functional cellular dysfunction of initial vasodilative dysfunction, EEG abnormality, metabolic derangement, and eventual irreversible structural injury with cellular death and imaging abnormality [15,16]. Mitigating the vasoconstrictive or vasodilative neurovascular coupling during intense ischemic depolarization may provide a novel hemodynamic mechanism of neuroprotection in accordance with the core–penumbra concept [26]. Therefore, it may be feasible to identify whether there is reversible or irreversible brain dysfunction in patients with comatose PCAS.

Our data suggest that the TCD-VMR is a useful real-time monitoring modality for applying emerging neuroprotectants or early outcome prediction at the bedside in the ICU. Cardiac arrest accounts for a high proportion of mechanically ventilated patients admitted to ICUs [1]. The length of ICU stays is increasing, but the rates of mortality and neurological morbidity due to anoxic brain injury remain very high. Sedatives used during TTM and post-cardiac arrest organ dysfunction can induce temporary brain dysfunction, which places a burden on caregivers and families with an unacceptably long period of uncertainty after the introduction of standard treatment for patients with PCAS. Therefore, early and accurate prognostication methods need to be established, both to avoid prolonged treatment of patients where continued life-supporting measures are futile and to ensure that patients with potential for recovery receive optimal management [4].

TCD-VMR as a semi-quantitative CA measurement can be affected by age, medical history (hypertension, diabetes, etc.), and cerebral vasculature. For these reasons, VMR may not always reflect the consequences of an acute brain insult. Even after we adjusted for the aforementioned variables in our data, VMR still remained an independent predictor of prognostic outcome in the final multivariate model. Therefore, TCD-VMR testing can be a useful technique to prognosticate the outcome in patients who have acute cerebral damage.

Our TCD-VMR test has some strengths; it is an intuitive and vasoregulatory reserve that is easily measured even without experts in the field. It can show the value of automated VMR using a blinded approach to minimize self-fulfilling prophecy. Unlike EEG prognostication, it is rarely affected by muscle artefacts or ICU environment regardless of the presence or absence of an EEG expert. In addition, due to the portability, noninvasive nature, and highly temporal resolution of TCD, it can be performed at bedside even in emergency situations and applied to daily clinical practices [27]. However, we encountered some limitations. Firstly, our data should be cautiously interpreted owing to the single-center design that may introduce bias, even if we analyzed a relatively homogeneous cohort of patients with comatose PCAS. Secondly, there are some technical limitations of the TCD procedure considering the relatively long learning curve for operator-dependent insonation of the MCA flow, poor temporal window issue, and probe fixation or insonation angle. As a result, TCD information was not obtained in approximately 12% of all participants given these technical limitations. Thus, novel technologies to simply detect the impaired CA function at bedside are needed in the future. Lastly, this study did not show treatment based on early evaluated CA by TCD-VMR. In future studies, additional research may be needed to determine the effect of treatment as a function of CA measured with TCD-VMR on the prognosis of patients with comatose PCAS.

## 5. Conclusions

In conclusion, our data suggest that TCD-VMR test is a useful tool at bedside to explore the outcomes of patients with comatose PCAS during the TTM, as well as regardless of sedation.

## Figures and Tables

**Figure 1 jcm-10-03386-f001:**
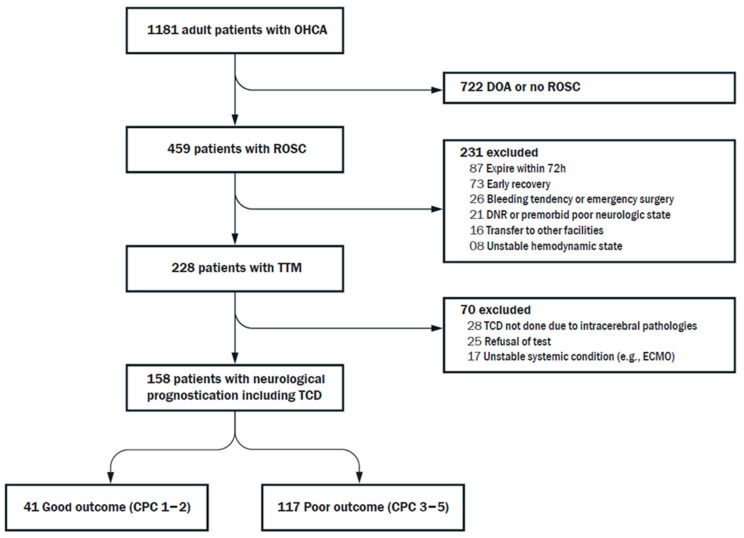
Flow chart. CPC, cerebral performance category; DOA, death on arrival; ECMO, extracorporeal membrane oxygenation; OHCA, out-of-hospital cardiac arrest; ROSC, return of spontaneous circulation; TCD, transcranial Doppler; TTM, targeted temperature management.

**Figure 2 jcm-10-03386-f002:**
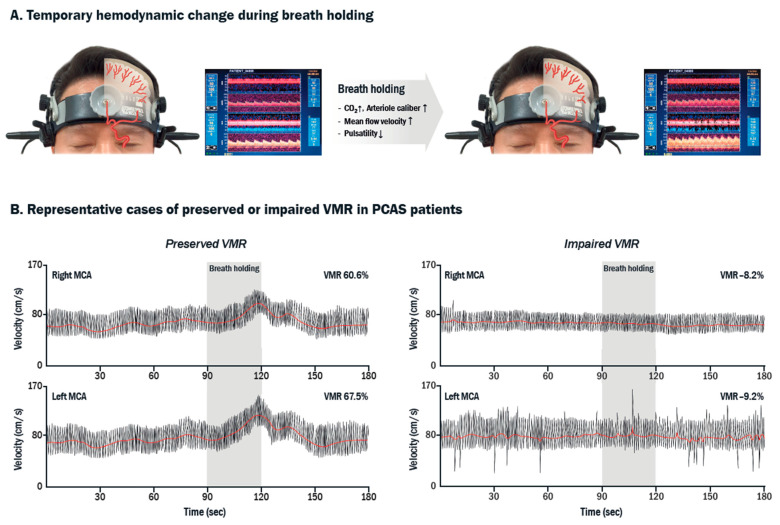
Methods of vasomotor reactivity test using transcranial Doppler (TCD-VMR). (**A**) Temporary hemodynamic change during TCD-VMR test by breath-holding maneuver. (**B**) Representative cases of preserved or impaired vasomotor reactivity in patients with post-cardiac arrest syndrome.

**Figure 3 jcm-10-03386-f003:**
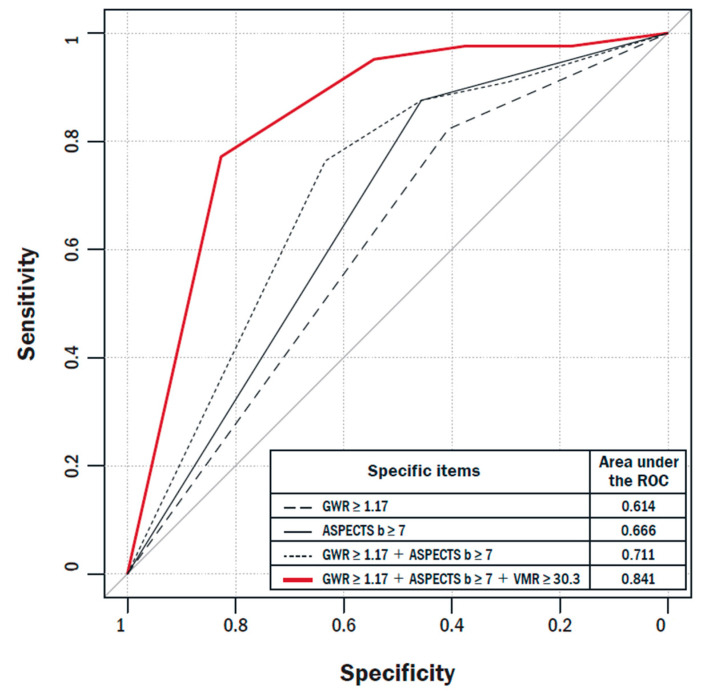
Receiver operating characteristic curve compared by adding vasomotor reactivity parameters to the traditional imaging study. ASPECTS-b, bilateral Alberta stroke program early computed tomography score; GWR, gray-to-white matter ratio; VMR, vasomotor reactivity.

**Table 1 jcm-10-03386-t001:** Baseline characteristics and prognostic parameter according to CPC.

	Overall (*n* = 158)	Good CPC (*n* = 41)	Poor CPC (*n* = 117)	*p*-Value
Age, years	60 (48–72)	56 (41–62)	61 (51–73)	0.015
Sex, male, *n* (%)	103 (65.2)	32 (78.1)	71 (60.7)	0.045
Witness arrest, *n* (%)	109 (69.0)	31 (75.6)	78 (66.7)	0.287
Bystander CPR, *n* (%)	114 (72.2)	29 (70.7)	85 (72.65)	0.814
Time to ROSC, min	25 (15–38)	15 (11–26)	26 (17–40)	<0.001
Initial rhythm, *n* (%)				<0.001
Shockable	41 (26.0)	28 (68.3)	13 (11.1)	
Non-shockable	117 (74.1)	13 (31.7)	104 (88.9)	
Cause of cardiac arrest, *n* (%)				<0.001
Cardiogenic	60 (38.0)	31 (75.6)	29 (24.8)	
Other medical	44 (27.8)	4 (9.8)	40 (34.2)	
Asphyxia	41 (26.0)	4 (9.8)	37 (31.6)	
Other	13 (8.2)	2 (4.9)	11 (9.4)	
Comorbidities, *n* (%)				
Hypertension	63 (39.9)	15 (36.6)	48 (41.0)	0.617
Diabetes	48 (30.4)	11 (26.8)	37 (31.6)	0.566
Cardiac diseases	31 (19.6)	10 (24.4)	21 (18.0)	0.371
Clinical parameters, *n* (%)				
No pupillary light reflex bilaterally	68 (43.0)	3 (7.3)	65 (55.6)	<0.001
GCS motor score	1.8 ± 1.3	2.6 ± 1.8	1.5 ± 1.0	<0.001
GCS motor score ≥2	128 (81.0)	16 (39.0)	14 (12.0)	<0.001
Serologic marker				
Initial S100, µg/L	3.04 (1.23–5.89)	0.94 (0.46–2.58)	3.85 (2.13–6.74)	<0.001
24 h S100, µg/L	0.76 (0.16–5.21)	0.11 (0.07–0.18)	2.24 (0.46–7.38)	<0.001
Imaging parameters				
Time from ROSC to CT, h	1.6 (1.0–2.3)	1.40 (1.0–2.1)	1.7 (1.0–2.6)	0.579
GWR	1.18 ± 0.08	1.23 ± 0.06	1.16 ± 0.08	<0.001
ASPECTS-b, points	7 (2–14)	14 (8–18)	5 (2–11)	<0.001
TCD parameters				
Time from ROSC to TCD, h	29.7 (23.0–46.6)	31.90 (22.7–42.1)	28.8 (23.1–46.9)	0.852
Bilateral poor temporal window, *n* (%)	14 (8.9)	2 (4.9)	12 (10.3)	0.223
Baseline mean flow velocity, cm/s	63.4 ± 27.1	51.9 ± 21.1	69.4 ± 28.1	0.001
Baseline mean pulsatility index	0.8 ± 0.3	0.8 ± 0.3	0.9 ± 0.3	0.704
Mean VMR, %	35.1 ± 37.4	54.4 ± 33.0	25.1 ± 35.8	<0.001
Reverberating flow, *n* (%)	29 (18.4)	0 (0.00)	29 (24.8)	<0.001
Electrophysiological parameters after TTM, *n* (%)				
No EEG reactivity (*n* = 146)	126/146 (86.3)	18/32 (56.3)	108/114 (94.7)	<0.001
Absence of bilateral N20 (*n* = 123)	67/123 (54.5)	0/32 (0.0)	67/91 (73.6)	<0.001

Values are presented as medians (interquartile range), means ± standard deviation, or numbers (percentages). Abbreviations: ASPECTS-b, bilateral Alberta stroke program early CT score; CPC, cerebral performance category; CPR, cardiopulmonary resuscitation; CT, computed tomography; EEG, electroencephalography; GCS, Glasgow Coma Scale; GWR, gray-to-white matter ratio; ROSC, return of spontaneous circulation; TCD, transcranial Doppler; TTM, targeted temperature management; VMR, vasomotor reactivity.

**Table 2 jcm-10-03386-t002:** Logistic regression analyses for predicting good CPC within 72 h.

Variables	Univariate Analysis	*p*-Value	Multivariable Analysis	*p*-Value
	Odds Ratio (95% CI)		Odds Ratio (95% CI)	
Baseline demographics				
Age (per 1 year increase)	0.97 (0.95–1.00)	0.020	0.93 (0.87–0.98)	0.009
Shockable rhythm	17.23 (7.19–41.32)	<0.001	14.17 (2.00–100.41)	0.008
Cardiogenic cause	9.41 (4.11–21.51)	<0.001	12.80 (1.81–90.42)	0.011
Time to ROSC (per 1 min increase)	0.95 (0.92–0.98)	<0.001	1.01 (0.95–1.07)	0.723
Early prognostic parameters				
Presence of pupillary light reflex	15.83 (4.63–54.21)	<0.001	2.16 (0.28–16.65)	0.459
GCS motor score ≥2	4.71 (2.03–10.91)	<0.001	1.10 (0.16–7.66)	0.926
GWR ≥1.17	6.07 (2.49–14.81)	<0.001	15.74 (1.11–223.89)	0.042
ASPECTS-b ≥7	10.35 (3.79–28.29)	<0.001	9.32 (1.07–81.26)	0.043
Initial S100 <2.37	6.18 (2.78–13.73)	<0.001	1.06 (0.24–4.77)	0.940
Mean VMR ≥30.3%	17.42 (4.92–61.63)	<0.001	122.80 (8.66–1741.91)	<0.001

Abbreviations: ASPECTS-b, bilateral Alberta stroke program early computed tomography score; CI, confidential interval; CPC, cerebral performance category; GCS, Glasgow Coma Scale; GWR, gray-to-white matter ratio; ROSC, return of spontaneous circulation; VMR, vasomotor reactivity.

**Table 3 jcm-10-03386-t003:** Comparisons of AUCs, IDI, and NRI for predicting good CPC at 1 month using CT methods and TCD-VMR test.

	AUC	Difference in AUC	Standard Error of Difference in AUC	95% CI of Difference in AUC	*p*-Value (Difference in AUCs)	Relative IDI (%)	*p*-Value (Relative IDI)	NRI	*p*-Value (NRI)	Valid *n*
GWR ≥ 1.17	0.614	0.205	0.043	0.121–0.289	<0.001	Ref.		Ref.		115
GWR ≥ 1.17 + VMR ≥ 30.3	0.819					0.28	<0.001	0.36	0.001	115
ASPECTS-b ≥ 7	0.666	0.170	0.041	0.090–0.250	<0.001	Ref.		Ref.		115
ASPECTS-b ≥ 7 + VMR ≥ 30.3	0.837					0.24	<0.001	0.25	0.018	115
GWR ≥ 1.17 + ASPECTS-b ≥ 7	0.711	0.131	0.040	0.052–0.209	0.001	Ref.		Ref.		115
GWR ≥ 1.17 + ASPECTS-b ≥ 7 + VMR ≥ 30.3	0.841					0.22	<0.001	0.20	0.041	115

Abbreviations: ASPECTS-b, bilateral Alberta stroke program early computed tomography score; AUC, area under the receiver operating characteristics curve; CI, confidential interval; CPC, cerebral performance category; GWR, gray-to-white matter ratio; IDI, integrated discrimination improvement; NRI, net reclassification improvement; TCD, transcranial Doppler; VMR, vasomotor reactivity.

## Data Availability

The data that support the findings of this study are available upon reasonable request to the corresponding author.

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
