# Peer review of "Prognostic Effects of Vasomotor Reactivity during Targeted Temperature Management in Post-Cardiac Arrest Patients: A Retrospective Observational Study"

_jcm, 2021, doi:10.3390/jcm10153386_

Round 1

Reviewer 1 Report

The topic presented in this manuscript is indeed noteworthy and should be further investigated. The article does have a substantial citation potential.

Main points (in the "chronologic" order of the manuscript)

Methods - 2.4 TCD-VMR

Were the sonographers certified – if Yes add the certifying society. There is a lack of reference for  performing TCD and breath-holding index.

Results

There is a lot of statistically significant differences in patients characteristics as well as in table 2. Author should mention that significancies in the text

Discussion

Main concern – as presented in results there are of statistically significant parameters predicting CPC at 1 month – not only VMR. That finding should be also discussed and explanation why to Choose VMR as a predicting factor should be mentioned.

Line 231 – increase in VMR … I would rather say decreased VMR is related to poor outcomes. Could You find cite any reference values of VMR or values obtained by other authors.

Add some comments for the possibility of the use of VMR in daily clinical practice.

Conclusions

The second sentence .. In future studies, … is not a conclusion . Please, change it.

It has been a pleasure to review this important piece of work.

Reviewer 2 Report

Thank you for the opportunity to review this interesting manuscript. Choi et al.  showed the prognostic effect of vasomotor reactivity during TTM in post- CA patients.

Overall, the manuscript is well written, the analyses are thoughtfully conducted, and the results are clinically quite interesting. I have no major concerns or comments, just some minor suggestions the authors might consider to further strengthen their presentation.

Greater interest is in the prognostication of good outcome.

  1. I suggest adding the contingency table data (true positive, false positive, true negative, and false negative) in order to perform a sensitivity analysis for good outcome prediction (sensitivity, specificity, and predictive values). If possible starting with 100% sensitivity for good outcome and then going down, specifying the cut-off values for continuous variables, such as GWR, ASPECTs, S100B protein, VMR. A good example that can help you might be Scarpino Resuscitation 2021 “SSEP amplitude accurately predicts both good and poor neurological outcome early after cardiac arrest: a post-hoc analysis of the ProNeCA multicentre study”.

  1. You did not report the overall GCS: I suggest to include this parameter in the study results, and if possible specify a range of motor response (MR) for predicting good outcome. M≥2 is not sufficient to know what the neurological status is at the time of the predictor evaluation. Since you define all patients included “comatose”, can you specify what the range of MR is? Were the patients out of sedation or not?

  1. Causes of death (mRS score 6, CPC score 5, or KOSCHI score 0) should be reported, specifically as cardiovascular or neurological as the determining factor (see table 2 of Geocadin et al. 2019).

  1. Interestingly, approximately 18% of included patients had a reverberant flow, a sign of cerebral hypertension with impending brain death. Can you tell how many patients were brain dead?

  1. You have reported old Guidelines: replace them with newer ones ( Nolan JP, Sandroni C, Böttiger BW, Cariou A, Cronberg T, Friberg H, Genbrugge C, Haywood K, Lilja G, Moulaert VRM, Nikolaou N, Olasveengen TM, Skrifvars MB, Taccone F, Soar J. European Resuscitation Council and European Society of Intensive Care Medicine guidelines 2021: post-resuscitation care. Intensive Care Med. 2021 Apr;47(4):369-421. doi: 10.1007/s00134-021-06368-4. Epub 2021 Mar 25. PMID: 33765189).

Round 2

Reviewer 1 Report

I accept changes made by authors in response to my previous comments.

Now, the manuscript is suitable for pubication

Reviewer 2 Report

The authors made the required changes to the text and tables, adding all the information I requested.